# Rhamnose Binding Protein as an Anti-Bacterial Agent—Targeting Biofilm of *Pseudomonas aeruginosa*

**DOI:** 10.3390/md17060355

**Published:** 2019-06-14

**Authors:** Tse-Kai Fu, Sim-Kun Ng, Yi-En Chen, Yuan-Chuan Lee, Fruzsina Demeter, Mihály Herczeg, Anikó Borbás, Cheng-Hsun Chiu, Chung-Yu Lan, Chyi-Liang Chen, Margaret Dah-Tsyr Chang

**Affiliations:** 1Institute of Molecular and Cellular Biology, National Tsing Hua University, Hsinchu 30013, Taiwan; gn00042366@gmail.com (T.-K.F.); flyingsim02@hotmail.com (S.-K.N.); ian8004@gmail.com (Y.-E.C.); yclee@jhu.edu (Y.-C.L.); cylan@life.nthu.edu.tw (C.-Y.L.); 2Simpson Biotech Co., Ltd., Taoyuan 333, Taiwan; 3Department of Biology, Johns Hopkins University, Baltimore, ML 21218, USA; 4Department of Pharmaceutical Chemistry, University of Debrecen, Debrecen 4032, Hungary demeter.fruzsina@science.unideb.hu (F.D.); herczeg.mihaly@science.unideb.hu (M.H.); borbas.aniko@pharm.unideb.hu (A.B.); 5Department of Life Science, National Tsing Hua University, Hsinchu 30013, Taiwan; chchiu@adm.cgmh.org.tw; 6Molecular Infectious Disease Research Center, Chang Gung Memorial Hospital, Chang Gung University College of Medicine, Taoyuan 333, Taiwan; dinoschen@adm.cgmh.org.tw

**Keywords:** rhamnose-binding protein, anti-biofilm, quorum sensing factor, anti-infection

## Abstract

More than 80% of infectious bacteria form biofilm, which is a bacterial cell community surrounded by secreted polysaccharides, proteins and glycolipids. Such bacterial superstructure increases resistance to antimicrobials and host defenses. Thus, to control these biofilm-forming pathogenic bacteria requires antimicrobial agents with novel mechanisms or properties. *Pseudomonas aeruginosa*, a Gram-negative opportunistic nosocomial pathogen, is a model strain to study biofilm development and correlation between biofilm formation and infection. In this study, a recombinant hemolymph plasma lectin (rHPL_OE_) cloned from Taiwanese *Tachypleus tridentatus* was expressed in an *Escherichia coli* system. This rHPL_OE_ was shown to have the following properties: (1) Binding to *P. aeruginosa* PA14 biofilm through a unique molecular interaction with rhamnose-containing moieties on bacteria, leading to reduction of extracellular di-rhamnolipid (a biofilm regulator); (2) decreasing downstream quorum sensing factors, and inhibiting biofilm formation; (3) dispersing the mature biofilm of *P. aeruginosa* PA14 to improve the efficacies of antibiotics; (4) reducing *P. aeruginosa* PA14 cytotoxicity to human lung epithelial cells in vitro and (5) inhibiting *P. aeruginosa* PA14 infection of zebrafish embryos in vivo. Taken together, rHPL_OE_ serves as an anti-biofilm agent with a novel mechanism of recognizing rhamnose moieties in lipopolysaccharides, di-rhamnolipid and structural polysaccharides (Psl) in biofilms. Thus rHPL_OE_ links glycan-recognition to novel anti-biofilm strategies against pathogenic bacteria.

## 1. Introduction

Part of the reasons for bacterial resistance to antibiotics are due to the formation of biofilm, composed of secreted polysaccharides, proteins, glycolipids and small molecules in the bacterial microenvironment. Recent development of antimicrobial agents with novel molecular mechanisms to control bacterial infectious diseases has drawn much attention [1]. The inhibition of bacterial biofilm formation by non-microbicidal mechanisms is an example of anti-pathogenic approaches [2,3]. One nosocomial infectious bacteria, *Pseudomonas aeruginosa*, is responsible for various infections, particularly in immuno-compromised individuals [4], and it forms biofilms to make antibiotic treatments inefficient and therefore promotes acute infections, for example, at upper airway, skin, wound, urinary tract, enteric and lung [5]. *P. aeruginosa* PA14 is a clinically isolated and highly virulent strain representing the most common clonal group worldwide [6]. The PA14 genome showing a high degree of conservation compared to that of the *P. aeruginosa* strain PAO1 contains two specific pathogenicity islands. The PA14 islands carried several genes implicated in virulence that are absent in PAO1, including genes encoding effectors of the type III secretion system for secreting virulent factors [7,8]. These characteristics make PA14 a threat to public health, but a good model for *P. aeruginosa* infection studies.

Biofilm development is associated with changes in bacteria phenotype and metabolic pathways [9]. During biofilm development, physiological changes of bacterial cells are regulated by a chemical signaling mechanism involving cell-to-cell communication, such as quorum-sensing (QS) signaling. The process of *P. aeruginosa* biofilm development is mainly regulated by three interconnected QS systems: Two use acyl-l-homoserine lactones (AHLs) and the third uses aquinolone [10]. In the AHL QS system, LasI-synthase generates *N*-(3-oxododecanoyl)-l-homoserine lactone [11] and RhlI synthase generates *N*-butyryl-l-homoserine lactone [12]. The former is a key factor in the maturation stage of the biofilm [13], and the latter is responsible for production of another biofilm regulator, di-rhamnolipid [14]. Numerous molecules with anti-QS property have been reported to inhibit biofilm formation of *P. aeruginosa* [15]. AHL analogues have inhibitory activities on biofilm formation in *P. aeruginosa* by down-regulating LasR-based QS system (LasR, a transcriptional regulator response for AHL). Some modified AHL analogues can also down-regulate pyocyanin, a virulence factor with elastase activity [16]. Certain enzymes are secreted by mammalian cells such as paraoxonases, which lactonase activity can degrade *P. aeruginosa* AHLs and further inhibit QS and biofilm formation. Most of these AHL pathway inhibitors efficiently work under 10 μM.

There already exist many different anti-biofilm agents from various resources. For example, cultured broth from certain marine cold adapted bacteria destabilized biofilm of *Pseudomonas aeruginosa* [17] and some essential oils from Mediterranean plants or selected exopolysaccharide from marine bacteria acts as anti-QS factors to inhibit biofilm formation of *Pseudomonas aeruginosa* [18,19]. Although AHL pathway inhibitors already exist, they still cannot completely inhibit the biofilm produced by *P. aeruginosa* [15], indicating that biofilm development is not only controlled by the AHL pathway but also other pathways that might partially complement to develop biofilm. This hypothesis leads us to focus on factors that may directly regulate biofilm development. Di-rhamnolipid, as a glycolipid secreted by *P. aeruginosa*, has been implicated in each of the following phases of biofilm development: (i) Forming microcolonies [20], (ii) regulating both cell-to-cell and cell-to-surface interactions [21], (iii) creating and maintaining fluid channels for water and oxygen flow around the base of the biofilm [21] and (iv) facilitating 3-D mushroom-shaped structure formation [22]. Accordingly, di-rhamnolipid seems to be a regulator that might directly interact with biofilms. The influence of di-rhamnolipid to biofilm development was observed, but the interaction of di-rhamnolipid with other QS factors remains unclear. To develop anti-biofilm reagents directly targeting di-rhamnolipid might be a novel anti-biofilm approach.

Rhamnose binding proteins (RBPs) are mainly isolated from eggs, ovary cells of fish and invertebrates with l-rhamnose (Rha) or d-galactose binding specificities [23,24]. RBPs are typically located in immune-related tissues or cells [25], suggesting that RBPs may be relevant to self-defense mechanisms. RBPs may interact and agglutinate Gram-negative and Gram-positive bacteria by recognizing lipopolysaccharide (LPS) and lipoteichoic acid (LTA), respectively [26]. RBPs can recognize some *O*-antigens and bind to glycolipids and glycoproteins of fish pathogens [27]. The RBP receptor is expressed on peritoneal macrophages of fish after an inflammatory stimulation [28]. The tissue specificity of expression and ability to interact with bacteria indicates that RBPs are naturally related to the innate immune system in animals as a pathogen recognition element.

An RBP possessing specific binding of l-rhamnose was discovered from horseshoe crab plasma and recently engineered for expression in the *Escherichia coli* system [29]. This recombinant horseshoe crab plasma lectin (rHPL) possesses a very low sequence identity with known RBPs and does not have conserved domains. Interestingly, rHPL binds to bacteria or pathogen-associated molecular patterns (PAMPs) by recognizing rhamnose moieties and inhibits the growth of *P. aeruginosa* [29]. Unlike other RBLs, rHPL only binds to l-rhamnose and rhamnobiose but not to galactose or mannose [30]. This high substrate specificity makes rHPL a prospective candidate to bind rhamnose-containing components in biofilm of *P. aeruginosa* and examine the biological functions of these bindings. 

## 2. Results

### 2.1. rHPL_OE_ Was Expressed in E. coli and Purified by Affinity Chromatography

rHPL was successfully expressed in *E. coli* in 2014. The yield of rHPL purified using a nickel-affinity column was ~8 mg/L, and the purity was 93% [29]. To improve the productivity and solubility of rHPL, the codon usage of synthetic *hpl* (*hploe*) was optimized for *E. coli*, and recombinant HPL_OE_ was co-expressed with chaperones. The expression product, rHPL_OE_, was purified by fast protein liquid chromatography (FPLC) equipped with a HisTrap™ affinity column following purification scheme in supplementary data showed in Appendix A. rHPL_DM_, representing rHPL with two artificial mutations Y88A and F145A, which showed no binding activities to a bacterial cell- (Appendix A) or pathogen-associated molecular pattern (PAMP; Appendix A), was applied as a negative control in this study. Appendix A illustrated the purification chromatogram and SDS polyacrylamide gel electrophoresis analysis data. The purified rHPL_OE_ was desalted and concentrated using an Amicon protein concentrator (10 kDa cut-off) and subjected to further analysis. Appendix A showed the secondary structure of rHPL_OE_ determined by circular dichroism (CD) spectroscopy in comparison to those of rHPL and rHPL_DM_, indicating that both rHPL_OE_ and rHPL_DM_ possessed similar structures to rHPL [29]. A yield of 11.33 mg/L rHPL_OE_ with a comparable purity. The yield of rHPL_OE_ was 1.4-fold higher than that of rHPL, suggesting that the chaperone improved the solubility and stability of rHPL in the *E. coli* system.

### 2.2. rHPL_OE_ Bound to Cell-Free Biofilm Matrix from P. aeruginosa PA14 via Recognizing Rhamnose

Our hypothesis is that rHPL_OE_ might bind to rhamnose-containing components in the biofilm and further interrupt bacterial biofilm development. First, binding capacity of rHPL_OE_ to the cell-free biofilm matrix from *P. aeruginosa* PA14 was tested. Here, a mature *P. aeruginosa* PA14 biofilm was extracted using a NaCl solution to give a cell-free biofilm matrix. In the extracted PA14 biofilm, the total protein was 0.19 mg/mL and polysaccharide, 0.32 mg/mL. Di-rhamnolipid, an important QS-factor and putative binding target of rHPL_OE_, was extracted using chloroform, and was 0.66 μg/mL by methylene blue method. The interaction between rHPL_OE_ and the cell-free biofilm matrix was measured by ELISA, and the inhibitory effect of monosaccharides and alginate on rHPL_OE_-biofilm interaction was determined using competitive ELISA. rHPL_OE_ at a final concentration of 0.1 μM was mixed with l-rhamnose and loaded into microplate wells coated separately with extracted *P. aeruginosa* PA14 biofilm. rHPL_OE_ mixed with buffer only was used as a positive control. rHPL_DM_ without rhamnose binding activity served as a negative control. As shown in Figure 1, addition of 25, 50, and 100 mM of l-rhamnose effectively reduced binding of rHPL_OE_ to the extracted biofilm, in comparison with the positive control. The inhibitory constant (*K_i_*) of l-rhamnose to rHPL_OE_ interaction with the biofilm of *P. aeruginosa* PA14 was 98 mM. The addition of 100 Mm of d-mannose, d-glucose, d-fructose, d-galactose and alginate could not reduce the binding between rHPL_OE_ and the extracted biofilm. These results indicated that rHPL_OE_ specifically bound to rhamnose containing components in *P. aeruginosa* PA14 biofilm.

### 2.3. Synthetic Rhamnobiosides and a Pentasaccharide Inhibited rHPL_OE_-Biofilm Interaction

To confirm that binding of rHPL_OE_ to the *P. aeruginosa* PA14 biofilm was via molecular recognition of rhamnose moiety, synthetic rhamno-oligosaccharides (Figure 2A,B) were used as competitors in a competitive ELISA. The chemical synthesis of phenylthio-rhamnobiosides and the Psl-pentasaccharide will be published elsewhere. The characterization of the new oligosaccharides used in this study is given in the Appendix A.

Figure 2 indicated that all synthetic compounds competed with the rHPL_OE_-biofilm interaction. The disaccharides are better inhibitor than monosaccharides, and (1-3)-linked rhamnosides were better than its (1-2)-linked counterparts. Psl-pentasaccharide showed the strongest competition presumably due to longer glycan chain length. The competitive effects of rhamnobiosides were stronger than those of rhamnobioses, indicating either a fixed anomeric configuration or hydrophobicity of aglycones slightly favors binding.

### 2.4. rHPL_OE_ Bound to Di-Rhamnolipid via Recognizing Rhamnose Portion

After confirming the binding activity of rHPL_OE_ to synthetic rhamnobiosides, direct binding activity of rHPL_OE_ to di-rhamnolipid from *P. aeruginosa* PA14 biofilm was measured by ELISA. Further competitive ELISA indicated that rHPL_OE_ binding to di-rhamnolipid could be competed more effectively by l-rhamnose compared to other monosaccharides (Figure 3).

### 2.5. rHPL_OE_ Inhibited Biofilm Formation and Dispersed the Preformed Biofilm of P. aeruginosa PA14

This experiment was conducted in M63 broth, a minimal low-osmolarity medium to slow down the growth rate of bacteria and mimic nutrient-depleted conditions for biofilm growth [31]. rHPL_DM_ was applied as a negative control. As shown in Figure 4A, 0.31 µM and higher concentrations up to 5 µM of rHPL_OE_ significantly inhibited the biofilm formation of *P. aeruginosa* PA14. Treatment with 0.31, 0.63, 1.25, 2.5 and 5 µM rHPL_OE_ significantly reduced biofilm formation to 49%, 51%, 39%, 18% and 13%, respectively. As expected, rHPL_DM_ did not inhibit biofilm formation. rHPL_OE_ was also evaluated for its biofilm dispersion activity with preformed biofilm. The mature biofilm of *P. aeruginosa* PA14 was dispersed by 16% and 24% upon treatment with 2.5 and 5 µM rHPL_OE_ for 24 h, respectively (Figure 4B). However, rHPL_DM_ did not disperse the mature biofilm. These results indicated that rHPL_OE_ inhibited biofilm formation and dispersed the mature biofilm of *P. aeruginosa* PA14 and that both activities perhaps are correlated with the rhamnose binding activity.

### 2.6. rHPLOE Inhibited Swarming Activity and Decreased Secreted Rhamnolipids of P. aeruginosa PA14

Swarming motility is positively correlated with the amount of extracellular rhamnolipids, leading to rapid bacterial translocation that promotes efficient colonization of bacterial cells on a surface [32]. Under optimal growth conditions, *P. aeruginosa* cell population would spread along a solid surface and cover a large area. If swarming were inhibited, the bacterial cell population would gather together to form a colony. The swarming area of *P. aeruginosa* PA14 was significantly reduced upon treatment with rHPL_OE_ but not rHPL_DM_ (Appendix A). The swarming area of PA14 was reduced by 33%, 43%, 48% and 46%, respectively, upon treatment with 0.63, 1.25, 2.5 and 5 µM rHPL_OE_ for 72 h (Appendix A). Figure 5 showed that the extracellular rhamnolipid levels of PA14 (11.7 ± 0.4 μg/mL in the control treated with buffer) reduced by 44%, 50%, 54%, 64% and 59% upon treatment with 0.31, 0.63, 1.25, 2.5 and 5 µM rHPL_OE_, respectively. However, the 5 µM rHPL_DM_ treatment did not affect the extracellular rhamnolipid levels of PA14. This result was consistent with the swarming assay, indicating that the binding of rHPL_OE_ to biofilm inhibited PA14 swarming activity by down-regulating the levels of extracellular di-rhamnolipid.

### 2.7. rHPL_OE_ Attenuated QS-Associated Factors

The development of biofilms appears to be regulated by the QS system. Most reagents that attenuate biofilm development work through inhibiting production of QS-factors or disrupting function of QS-factors by direct binding to QS-factors [33,34,35]. Among these agents, pyoverdine and pyocyanin are unique toxic molecules related to the virulence of *P. aeruginosa* PA14. To understand the attenuation mechanism of *P. aeruginosa* biofilm by rHPL_OE_, the effect of rHPL_OE_ on QS-mediated azocasein-degrading protease activity and the secretion of two virulent factors, pyoverdine and pyocyanin were evaluated. Bacterial extracellular proteases degrade proteins in host cells (infected tissue), thereby facilitating bacterial invasion and growth [36]. Interestingly, bacterial protease secretion by *P. aeruginosa* PA14 was down-regulated by rHPL_OE_ treatment (Appendix A). The azocasein-degrading assay showed that 1.25 and 2.5 µM rHPL_OE_ respectively decreased 16% and 12% of total proteolytic activities. Treatment with 0.31, 0.63, 1.25 and 2.5 µM rHPL_OE_ decreased the pyoverdine levels by 12%, 8%, 17% and 21%, respectively (Appendix A). Treatment with the same concentrations of rHPL_OE_ decreased the pyocyanin levels by 41%, 56%, 45% and 40%, respectively (Appendix A).

These results indicated that the binding of rHPL_OE_ to the biofilm down-regulated the protease activities and QS-factors (Table 1). Consistent with previous observations, 2.5 µM rHPL_DM_ did not show an inhibitory effect. Relatively weak inhibition of rHPL_OE_ on protease activity and pyoverdine expression indicated that these effects might be indirect effects. The effect of rHPL_OE_ on pyocyanin production was detectable but not concentration-dependent. A possible alternate reason for this may be that the inhibition effect of rHPL_OE_ on pyocyanin production was compensated by other regulatory pathways.

### 2.8. Combination Treatment with rHPL_OE_ Improved Efficacies of Antibiotics on P. aeruginosa with Preformed Biofilms

Since rHPL_OE_ exhibited dispersion of *P. aeruginosa* PA14 biofilm, we hypothesized that rHPL_OE_ might increase the activities of antibiotics against *P. aeruginosa* PA14 in mature biofilms. Total protein assay showed that the combination of rHPL_OE_ with either azithromycin (hydrophobic) or cephalexin (hydrophilic) significantly attenuated the total protein levels in the biofilm. The IC50 of azithromycin and cephalexin to *P. aeruginosa* PA14 with pre-formed biofilm was 27.3 ± 1.4 and 27.7 ± 0.9 μg/mL, respectively. In the presence of 25 μg/mL azithromycin, in combination with 0.31, 0.63, 1.25 and 2.5 µM rHPL_OE_ reduced the total protein in the biofilm by 19%, 21%, 39% and 43%, respectively, compared to the antibiotic-only control (Figure 6).

In the presence of 25 μg/mL cephalexin, combination treatment with the same concentrations of rHPL_OE_ reduced the total protein in biofilms by 33%, 33%, 38% and 50%, respectively, compared to the antibiotic-only control (Figure 6). This observation suggested that rHPL_OE_ might facilitate antibiotics to kill *P. aeruginosa* PA14 by partially interfering with biofilm regulation and thus destroying the structure of the mature biofilm.

### 2.9. rHPL_OE_ Inhibited Infection of P. aeruginosa in Mammalian Cells and Zebrafish Embryos

During *P. aeruginosa* infection, rhamnolipids and pyocyanin are the important virulence factors causing cytotoxicity. Since rHPL_OE_ inhibited rhamnolipid and pyocyanin production in *P. aeruginosa* PA14, we predicted that rHPL_OE_ could also reduce *P. aeruginosa* PA14 infection inhuman lung cells. To evaluate this hypothesis, cell death of human lung cell line A549 by *P. aeruginosa* PA14 infection in the presence of rHPL_OE_ was examined. As shown in Figure 7, the PA14-infected A549 cell death was reduced by 14%, 21%, 57% and 73% in the presence of 0.31, 0.63, 1.25 and 2.5 µM rHPL_OE_, respectively.

Regarding the negative control, 2.5 µM BSA or 2.5 µM rHPL_DM_ did not influence A549 cell death. These assays clearly demonstrated that the mortality rate of A549 cells killed by *P. aeruginosa* PA14 decreased due to the presence of rHPL_OE_. Therefore, rHPL_OE_ attenuated *P. aeruginosa* PA14 infection in human lung A549 cells through direct interaction between rHPL_OE_ and rhamnose-containing bacterial compounds.

Zebrafish (*Danio rerio*) has become an important vertebrate animal model for many disease studies including pathogen infection [37] in recent years. Here fertilized eggs from zebrafish were used as an animal model to evaluate whether rHPL_OE_ attenuated *P. aeruginosa* PA14 infection in vivo. Zebrafish embryos showed normal development in the absence of *P. aeruginosa* PA14, while the development of most embryos was significantly delayed or even died at 48 h following infection with *P. aeruginosa* PA14 (Figure 8).

However, when *P. aeruginosa* PA14 was pre-incubated with 5 μM rHPL_OE_, 82.2% of embryos developed normally as compared to control embryos showing a development rate of 90.9%. Zebrafish embryos treated with 2.5 μM rHPL_OE_ still developed slowly, but almost all of the embryos survived. These results indicated that rHPL_OE_ could protect zebrafish embryos from *P. aeruginosa* PA14 infection at concentrations higher than 2.5 μM.

## 3. Discussion

To study the carbohydrate components of the biofilm, carbohydrate-binding proteins, such as plant lectins, have been widely applied. Previously many different neutral carbohydrates including *N*-acetyl-d-glucosamine, *N*-acetyl-d-galactosamine, d-glucose and d-mannose were identified in the exopolymeric matrix of the biofilm [38]. Lectins with various specificities show an interaction with the carbohydrate components of biofilms. For example, concanavalin A from *Canavalia ensiformis* seeds is specific for d-glucose and d-mannose [39], and lectin from *Triticum vulgaris* germs is specific for *N*-acetyl-d-glucosamine and sialic acid [40]. l-Rhamnose as a unique sugar in bacteria and plants is a common component of the cell wall and capsule of many pathogenic bacteria including Gram-negative *P. aeruginosa* [41], *Salmonella typhimurium* [42], *Vibrio cholera* [43,44] as well as *Mycobacterium tuberculosis* [45]. *P*. *aeruginosa* produces at least three distinct exopolysaccharides that contribute to biofilm development and architecture: Alginate, Pel and Psl [46]. Alginate consists of only uronic acids, Pel is a glucose-rich polysaccharide and Psl consists of a repeating pentasaccharide containing d-mannose, d-glucose and l-rhamnose [47]. Psl serves as a primary structural scaffold for biofilm development. In addition, Psl is involved in early stages of biofilm formation and promoting cell-to-cell interactions [48,49]. Another rhamnose-containing component in the biofilm was the QS-factor di-rhamnolipid. A mutant strain (*P. aeruginosa* PAO1C-*rhlAB*) that could not produce rhamnolipids lost its swarming activity, which agreed with our observation in rHPL_OE_-treated PA14 [50]. rHPL_OE_ bound to both Psl and di-rhamnolipid through targeting rhamnose moiety and synthetic rhamnobiosides-competitive ELISA showed that rHPL_OE_ preferred to bind with α(1-3)-rhamnobiose rather than α(1-2)-rhamnobiose. The linkage of the rhamnose of Psl was reported to be α(1-3) [47] while that of di-rhamnolipid was α(1-2) [51], perhaps favoring that rHPL_OE_ binding to rhamnose component in biofilm was governed by recognition of unique structure feature of rhamnosyl moiety. Although rHPL_OE_ binding to Psl still required more abundant glycan for detail characterization, rHPL_OE_ binding to di-rhamnolipid down-regulated the expression of di-rhamnolipid and QS-factors of *P. aeruginosa* and further inhibited biofilm formation.

The most important finding of this study was that the binding of rHPL_OE_ to the *P. aeruginosa* PA14 biofilm inhibited biofilm development and disrupted the mature biofilm. Our data clearly showed that rHPL_OE_ caused these bio-effects by interacting with components in the biofilm including structural polysaccharides or di-rhamnolipid, largely due to the down-regulation of QS-factors including di-rhamnolipid, pyocyanin, pyoverdine and extracellular proteases by interrupting the functions of di-rhamnolipid.

Based on the biofilm dispersion activity of rHPL_OE_, we proposed that rHPL_OE_ possessed synergistic effects with antibiotics on *P. aeruginosa*. Two antibiotics commonly applied for treating *P. aeruginosa* infections, azithromycin, a hydrophobic azalide that kills bacteria by decreasing protein production and cephalexin, a hydrophilic beta-lactam that kills bacteria by inhibiting cell wall synthesis, were used in this study in combination with rHPL_OE_. The results showed that rHPL_OE_ significantly improved the bactericidal activity of both antibiotics, strongly suggesting that rHPL_OE_ was useful as a biofilm dispersion reagent for deconstructing the biofilm and improving the activities of antibiotics.

Many studies reported that *P. aeruginosa* infection could be inhibited by down-regulating QS-factors. Importantly, these anti-QS reagents (allicin, triazolyldihydrofuranone and baicalin hydrate) are effective against multidrug-resistant *P. aeruginosa* [33,34,35]. Studies in the past decade revealed that these anti-QS reagents also inhibited many human infections caused by biofilm-producing bacteria [52,53]. This fact is important for fighting human pathogenic bacteria, as biofilms are found to be involved in over 80% of microbial infections in humans [54]. Since rHPL_OE_ could reduce the levels of pyoverdine and pyocyanin, we speculate that rHPL_OE_ might also inhibit *P. aeruginosa* infection. It was found that rHPL_OE_ significantly reduced the cytotoxicity towards A549 cells and neutralized toxicity (leading to development retardation or death) of the zebrafish embryo caused by *P. aeruginosa* PA14.

## 4. Conclusions

Binding and down-regulation of di-rhamnolipid has not attracted much attention so far. In this study, we optimized the production of rHPL in *E. coli* using chaperone co-expression. rHPL_OE_, a highly specific rhamnose binding protein, bound to not only bacterial cells and PAMPs [29] but also extracted cell-free biofilm from *P. aeruginosa*. In addition, such interaction inhibited biofilm formation and dispersed mature biofilm through down-regulating secretion of di-rhamnolipid in biofilm and further down-regulating other QS-factors including extracellular proteases, pyoverdine and pyocyanin. As a biofilm dispersion reagent, rHPL_OE_ increased the antibiotic activity against *P. aeruginosa* PA14 with pre-formed biofilm. Therefore, rHPL_OE_ promised to be an effective anti-biofilm agent for combination therapy. At cellular and animal levels, rHPL_OE_ inhibited the infection and toxicity of *P. aeruginosa* PA14 towards human lung epithelial cells and zebrafish embryos. These results indicated that rHPL_OE_ served as a novel anti-biofilm agent by targeting rhamnose-containing components in biofilm, which in turn linked glycan-recognition to novel anti-biofilm strategies against pathogenic bacteria.

## 5. Materials and Methods 

### 5.1. Bacteria Strains, Growth Medium and Plasmid

*Escherichia coli* TOP10F’ (Invitrogen, Waltham, MA, USA) was used as a host for vector construction and DNA manipulation. *E. coli* BL21(DE3) (Invitrogen, Waltham, MA, USA) was used as a host for protein expression. *Pseudomonas aeruginosa* PA14 (serotype O19) was kindly provided by Dr. Hwan-You Chang (Institute of Molecular Medicine, National Tsing Hua University, Hsinchu, Taiwan). The vector pET-23a (+) (Novagen, Burlington, MA, USA) with a T7 promoter was used for recombinant protein expression in *E. coli* cells and sequence analysis. Takara’s Chaperone Plasmid Set (#3340, TaKaRa, Shiga, Japan) was used for chaperone co-expression. All other buffers and reagents were of the highest commercial purity.

### 5.2. Expression and Purification of rHPL_OE_

Chloramphenicol (≥98%, #C0378), ampicillin (96%–102%, #A9393), isopropyl β-d-1-thiogalactopyranoside (IPTG, ≥99%, #I6758), Tris base (ACS reagent, ≥99.8%, #252859), NaCl (ACS reagent, ≥99%, #746398), imidazole (≥99%, #I5513), HCl (#320331) and phenylmethylsulfonyl fluoride (PMSF, ≥99%, # 78830) was purchased from Sigma-Aldrich (St. Louis, MO, USA). Luria-Bertani (LB, #244610) was purchased from BD (Franklin lake, NJ, USA). rHPL_OE_ was the chaperone co-expressed product of the synthesized sequence (*hpl_OE_*) and the codon of *hpl_OE_* was optimized for *E. coli* expression. To generate a co-expression clone, pET23a-*hpl_OE_* was transformed into *E. coli* BL21(DE3) competent cells containing pG-KJE8 (#3340, TaKaRa, Shiga, Japan), and transformants were selected by an LB plate with 20 μg/mL chloramphenicol and 50 μg/mL ampicillin. After induction with a final concentration of 0.1 mM IPTG at 16 °C for 16 h, cells were harvested by centrifugation (KUBOTA, Osaka, Japan), and the residues were suspended in equilibrium buffer (20 mM Tris-HCl, 200 mM NaCl and 5 mM imidazole, pH 7.4) supplemented with a protease inhibitor (1 mM PMSF) and disrupted by three passages through a cell homogenizer (Sonicator 3000, Misonix, Farmingdale, NY, USA) at 15,000 psi. The recombinant proteins were purified using a HisTrap™ HP immobilized metal ion affinity column (#17524701, GE Healthcare, Little Chalfont, Buckinghamshire, UK) and ÄKTA start system (GE Healthcare, Little Chalfont, Buckinghamshire, UK). Purified proteins were then concentrated and buffer-exchanged to the Tris buffer (20 mM Tris-HCl and 200 mM NaCl, pH 7.4) using a 15 kDa cut off Amicon Ultra centrifugal filter unit (#UFC901096, Millipore, Burlington, MA, USA).

### 5.3. Biofilm Formation and Extraction

Bacterial biofilm extraction was as described by Chibaet et al. [55]. *P. aeruginosa* PA14 grown on LB plates was inoculated in LB medium and incubated overnight at 37 °C with 250 rpm circular shaking. A portion of the overnight culture was 1000-fold diluted in 10 mL LB medium and incubated at 37 °C to an A600nm of 1. Bacterial cells with biofilm were harvested from incubated solution by centrifugation at 8000× *g* for 10 min at 25 °C (KUBOTA, Osaka, Japan). The harvested pellet was re-suspended with 1 mL of 1.5 M NaCl to extract a cell-free biofilm component. The suspensions were centrifuged at 5000× *g* for 10 min at 25 °C (KUBOTA, Osaka, Japan), and the supernatants containing the biofilm fraction were collected.

### 5.4. Polysaccharide Quantification in the Extracted Biofilm

Absolute ethanol (≥99.8%, #32221), phenol (#P1037) and sulfuric acid (95%–98%, #435589) were purchased from Sigma-Aldrich (St. Louis, MO, USA). Polysaccharide in biofilm was measured as described by Tribedi and Sil [56]. Extracted biofilm solution was mixed with 2.2 volumes of chilled absolute ethanol, incubated at −20 °C for 1 h and centrifuged at 3500× *g* for 20 min at 4 °C (KUBOTA, Osaka, Japan). The pellet containing exopolysaccharides was dissolved in sterile water and measured by the phenol-sulfuric acid method. Fifty microliters of re-suspended sample was mixed with 200 μL phenol, and then, 750 μL sulfuric acid was added. The solution was left standing for 40 min, and vigorously shaken. After shaking, the A_490nm_ was measured using a spectrophotometer (U3310, HITACHI, Tokyo, Japan), and the amount of total sugar (excluding amino sugars) was calculated with a standard curve established using glucose.

### 5.5. Binding Activity of rHPL_OE_ to Biofilm or Di-Rhamnolipid

Di-rhamnolipid (95%, #R95DD), phosphate buffer saline (PBS, #P3813), Tween-20 (#P1379), l-rhamnose (≥99%, #R3875), d-glucose (≥99.5%, #G8270), d-galactose (≥99%, #G0750), d-fructose (≥99%, #F0127), d-mannose (≥99%, #M8574) and sodium alginate (#W201502) was purchased from Sigma-Aldrich (St. Louis, MO, USA). For competitive enzyme-linked immunosorbent assays (ELISA), the extracted biofilm or prepared di-rhamnolipid solution (120 μg/mL in ddH_2_O) was diluted in 10× the volume of the coating solution (#5150−0014, SeraCare KPL, Milford, MA, USA), and 50 μL of the mixture was added to each well of the 96-well microplate (#442404, Thermo Fisher, Waltham, MA, USA) and incubated at 4 °C overnight. After blocking with a blocking reagent (#10057177103, Roche, Basel, Switzerland) at 37 °C for 1 h, the plates were washed with PBST (PBS with 0.05% Tween-20) three times. To the washed wells, 25 μL of 0.2 μM rHPL_OE_ was mixed with 25 μL of two-fold the indicated concentration of l-rhamnose, d-glucose, d-galactose, d-fructose, d-mannose or sodium alginate and the mixture was added to the wells, which were maintained at 37 °C for 1 h. Fifty microliters of 0.1 μM purified rHPL_OE_ was added in parallel as a positive control, and Tris-buffer (20 mM Tris-HCl and 200 mM NaCl, pH 7.4) was added in parallel as a negative control. After washing three times with PBST, the microplates were incubated with monoclonal anti-His-tag (1:5000; #631212, Clontech, Mountain View, CA, USA) in PBST at 37 °C for 1 h. Subsequently, horseradish peroxidase-conjugated anti-mouse IgG (1:5000; #AB2338512, Jackson ImmunoResearch, West Grove, PA, USA) in PBST was added to the microplates, and after washing three times with PBST, the plates were incubated at 37 °C for 1 h. After being washed three times with PBST, 100 μL of 3,3’,5,5’-tetramethylbenzidine substrate (#5120-0077, SeraCare KPL, Milford, MA, USA) was added to each well and incubated at 37 °C for exactly 15 min. Finally, the reaction was terminated by the addition of 100 μL of 2 N H_2_SO_4_. The A_450nm_ was read using a microplate spectrophotometer (iMark Microplate Absorbance Reader, Bio-Rad, Hercules, CA, USA). The inhibitory constant (*K_i_*) was analyzed with GraphPad 5.0 software (GraphPad Software, La Jolla, CA, USA; fitted with a Binding-Competitive/Onesite-Fit *Ki*).

### 5.6. Anti-Biofilm Assay

KH_2_PO_4_ (≥99%, #P5655), K_2_HPO_4_ (≥99%, #P3768), (NH_4_)_2_SO_4_ (≥99%, #A4418), MgSO_4_ (≥99.5%, #M7506), L-arginine (≥98%, #A5006), polymyxin B (meets USP testing specifications, #P0972) and crystal violet (≥90%, #C0775) was purchased from Sigma-Aldrich (St. Louis, MO, USA). Biofilm formation was described by O’Toole [57]. Briefly, overnight culture of the *P. aeruginosa* PA14 in LB medium was diluted with fresh M63 medium (22 mM KH_2_PO_4_, 40 mM K_2_HPO_4_, 15 mM (NH_4_)_2_SO_4_, 1 mM MgSO_4_ and 0.4% l-arginine). A total of 1 × 10^6^ CFU/well bacterial cells were then cultured with various concentrations of rHPL_OE_ in a final volume of 100 µL M63 medium onto a 96-well clear round bottom polystyrene microplate (#3788, Thermo Fisher, Waltham, MA, USA) and incubated at 37 °C for 24 h without shaking. A standard antibiotic polymyxin B (50 µg/mL) often used to kill *P. aeruginosa* in other studies was applied as a positive control, and M63 only was used as a negative control. The suspended culture was then discarded, and the plate was washed with ddH_2_O twice to remove any remaining suspended cells in the microtiter wells. The biofilm was then stained with 125 µL of 0.1% crystal violet for 15 min, after which the stained biofilm was washed with ddH_2_O five times to remove any unbound dye. The crystal violet bound to the biofilm was solubilized with 125 µL 30% (*v*/*v*) acetic acid and quantified by measuring the A_595nm_ using the microplate spectrophotometer (iMark Microplate Absorbance Reader, Bio-Rad, Hercules, CA, USA). For the biofilm dispersion test, biofilm cultures were grown statically in 80 µL of M63 medium for 24 h, followed by the addition of 20 µL of the indicated concentration of rHPL_OE_ in M63 medium and incubation for an additional 24 h. The addition of polymyxin B (50 µg/mL) to the culture before biofilm formation was used as a positive control.

### 5.7. Swarming Motility Measurement 

The swarming motility of *P. aeruginosa* PA14 was investigated in plates containing swarming motility media (LB with 0.5% (wt) glucose and 0.6% (wt) agar). Agar was purchased from OXOID (#LP0011, Basingstoke, Hampshire, UK). An aliquot of 2 μL of motility media containing 1 × 10^6^ CFU/mL bacterial cells was inoculated in plates with different concentrations of rHPL_OE_. Subsequently, spots were dired for 20 min at room temperature and incubated at 37 °C for 48 h. The diameter of the circular bacterial growth was measured [50].

### 5.8. Extracellular Rhamnolipid Quantification

Methylene blue (≥82%, #M9140), sodium borate (ACS reagent, ≥99.5%, #S9640) and chloroform (≥99.5%, #C2432) was purchased from Sigma-Aldrich (St. Louis, MO, USA). Rhamnolipid was quantified as described by Pinzon et al. [58]. The supernatant of the anti-biofilm test was collected, and bacterial cells were removed by centrifugation (KUBOTA, Osaka, Japan). The pH of the cell-free supernatant was first adjusted to 2.3 ± 0.2 using 1 N HCl. Two hundred microliters of acidified sample was then extracted with the same volume of chloroform five times. One milliliter of the chloroform extract was carefully removed and mixed with 40 μL of 0.1% methylene blue (freshly prepared and pH adjusted to 8.6 ± 0.2 with 50 mM borate buffer) and 960 μL of distilled water in a 2 mL tube. After being vigorously shaken for 4 min, the samples were left to stand for 15 min. The chloroform phase was transferred into a quartz cuvette and value of A_638 nm_ was measured by spectrophotometer (U3310, HITACHI, Tokyo, Japan). The values were converted to rhamnolipid concentrations using a calibration curve established by applying the same procedure to standard rhamnolipid solutions of different concentrations.

### 5.9. Azocasein Degradation Assay

The proteolytic activity was determined in the cell-free culture supernatant according to the method by Kessler et al. [59]. The amount of protease released by *P. aeruginosa* treated by rHPL_OE_, the supernatant of the anti-biofilm test was collected by centrifugation (KUBOTA, Osaka, Japan). One hundred and fifty microliters of supernatant were added to 0.3% of 1 mL azocasein (#A2765, Sigma-Aldrich St. Louis, MO, USA) in 50 mM Tris-HCl (pH 7.5) and incubated at 37 °C for 15 min. The reaction was stopped by the addition of 0.5 mL 10% trichloroacetic acid, and the mixture was centrifuged at 8000× *g* for 5 min (KUBOTA, Osaka, Japan) to obtain a clear supernatant. The absorbance of the clear supernatant was then measured at 400 nm (U3310, HITACHI, Tokyo, Japan) [60].

### 5.10. Pyoverdine Quantification Assay

*P. aeruginosa* was incubated with different concentrations of rHPL_OE_ at 37 °C for 24 h. Thereafter, the cultured liquid was centrifuged at 8000× *g* for 15 min (KUBOTA, Osaka, Japan), and the cell-free supernatant was used for the pyoverdine measurement. The relative concentrations of pyoverdine in all of the treated supernatants with respect to the control was measured through a fluorescence microplate spectrophotometer (Victor2, PerkinElmer, Waltham, MA, USA) at an excitation wavelength of 405 nm and an emission wavelength of 465 nm [61].

### 5.11. Pyocyanin Quantification Assay

The pyocyanin quantification assay was performed according to the method described by Essar et al. [62]. *P. aeruginosa* was incubated with different concentrations of rHPL_OE_ at 37 °C for 24 h, and the cell-free supernatant was collected by centrifugation (KUBOTA, Osaka, Japan). Five milliliters of the supernatant were extracted with 3 mL of chloroform and the chloroform layer was re-extracted with 1 mL of 0.2 N HCl to produce an orange yellow to pink solution, and the chloroform phase was transferred into a quartz cuvette and value of A_520 nm_ was measured by a spectrophotometer (U3310, HITACHI, Tokyo, Japan).

### 5.12. Synergistic Effect of rHPL_OE_ with Antibiotics 

The experiment design was according to the method described by Das et al. [63]. *P. aeruginosa* with a mature biofilm was treated with rHPL_OE_ in combination with either azithromycin (≥95%, #75199, Sigma-Aldrich, St. Louis, MO, USA) [63] or cephalexin (≥95%, #C4895, Sigma-Aldrich, St. Louis, MO, USA) [64]. Solutions of rHPL_OE_ in 0.3125 µM, 0.625 µM, 1.25 µM and 2.5 µM in combination with different concentrations of antibiotics were directly added to the cultured bacterial liquid with a mature biofilm and held at 37 °C for 24 h. The total protein and amount of biofilm were measured to validate the synergistic antibacterial effect of rHPL_OE_ and antibiotics.

### 5.13. Total Protein Concentration Measurement in the Biofilm

Concentration of the extractable protein was determined as a measure of the *P. aeruginosa* biofilm population density. The microbial population density in the biofilm is assumed to be directly proportional to the extractable protein concentration [56]. After incubation, planktonic *P. aeruginosa* cells were removed, and adhered cells remained in the biofilm were gently washed with sterile PBS and boiled for 30 min in 5 mL of 0.5 N NaOH (ACS reagent, ≥97%, #221465, Sigma-Aldrich, St. Louis, MO, USA) to extract the protein. After that, the suspension was centrifuged at 8000× *g* for 5 min (KUBOTA, Osaka, Japan), and the resulting clear supernatant was collected. The supernatant protein concentration was then measured by the bicinchoninic acid (BCA) protein assay kit (#23225, Thermo Fisher, Waltham, MA, USA).

### 5.14. Anti-A549 Infection Assay

The Roswell Park Memorial Institute (RPMI)-1640 medium (#10-040 CMS) and Antibiotics–Antimycotic solution (PSA, #30-004-CIS) was purchased from Corning (Corning, NY, USA). Characterized fetal bovine serum (FBS, #SH30071) was purchased from HyClone Laboratory (Logan, UT, USA). The A549 infection model was modified from method described by Chi [65]. A549 cells (ATCC^®^ number: CCL-185™, BCRC, Hsinchu, Taiwan), adenocarcinomic human lung cells, were seeded in 96-well tissue culture plates at 2 × 10^4^ containing 100 µL of RPMI-1640 medium supplemented with 10% (*v*/*v*) fetal FBS and 1% (*v*/*v*) PSA and allowed to grow at 37 °C for 16 to 18 h. Culture supernatants were removed, the monolayer was washed once with PBS buffer, and then 50 µL of serum-free RPMI-1640 containing two times the indicated concentration of rHPL_OE_ was added to cells and incubated for 30 min, followed by inoculation with *P. aeruginosa* PA14. RPMI-1640 with 5 µM BSA or rHPL_DM_ was used as negative controls. For inoculation, the fresh bacterial cells cultured in LB broth were washed with PBS, re-suspended and diluted in RPMI-1640 medium to a concentration of 1 × 10^8^ CFU/mL. Thereafter, 50 µL of the bacterial dilution was applied to the rHPL_OE_-treated A549 cells at a multiplicity of infection (MOI) of 50 (i.e., 1 × 10^6^ CFU/50 µL/well). The blank group was A549 cells treated by 2.5 μM rHPL_OE_ without bacterial infection. After infection for 16 h at 37 °C, the A549 cell viability was determined by the AlamarBlue cell viability assay (BUF012B, Bio-Rad, Hercules, CA, USA). It should be noted that 50 µg/mL polymyxin B should be added to alamarBlue reagent to prevent a survival signal from residual bacteria. The level of viability, expressed as a percentage, was calculated as follows: % viability = [OD of assay cells/OD of control cells] × 100.

### 5.15. Anti-Zebrafish Infection Assay

Zebrafish embryos were purchased from Gendanio Biotech Inc. (New Taipei City, Taiwan). Healthy, transparent and regular embryos were selected and aliquoted into 96-well plates with four embryos per well, followed by incubation at 28 °C in 100 µL embryo water. Thereafter, 100 µL of *P. aeruginosa* (1 × 10^8^ CFU/mL) pre-incubated with 5 mM rHPL_OE_ for 1 h was added to the embryos at 24 h post-fertilization (hpf). The zebrafish embryos were further incubated at 28 °C, and the development of each embryo in 24 h and 48 h was observed using an inverted microscope (TS100, NIKON, Tokyo, Japan) equipped with a digital camera. The number of larva at 48 h in each group was counted, and the rate of successful development was calculated. All zebrafish related experiments were conducted in accordance with the ethical guidelines of Council of Agriculture, Executive Yuan and Ministry of Science and Technology.

### 5.16. Statistical Analyses 

All statistical analyses were carried out using GraphPad Prism version 5.01 for Windows (GraphPad Software, La Jolla, CA, USA). Each value was the average of three measurements, where the presented data is the mean ± SD. All means were compared by one-way ANOVA.

## Figures and Tables

**Figure 1 marinedrugs-17-00355-f001:**
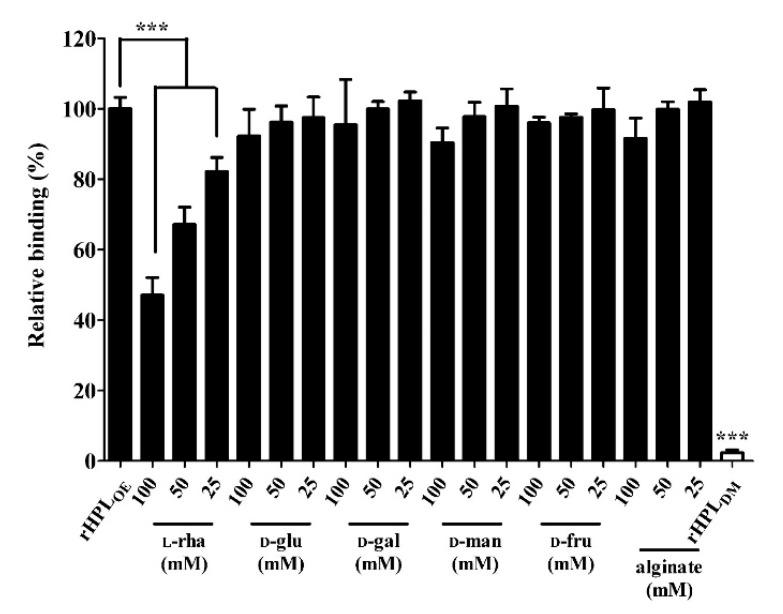
Inhibitory effects of monosaccharides on the recombinant hemolymph plasma lectin (rHPL_OE_)-biofilm interaction. The binding activity of rHPL_OE_ to the PA14 biofilm and inhibitory effects of monosaccharides or alginate on this binding were determined by competitive ELISA. rHPL_DM_ was applied as a negative control. *** *p* < 0.001 versus the rHPL_OE_ only group (positive control).

**Figure 2 marinedrugs-17-00355-f002:**
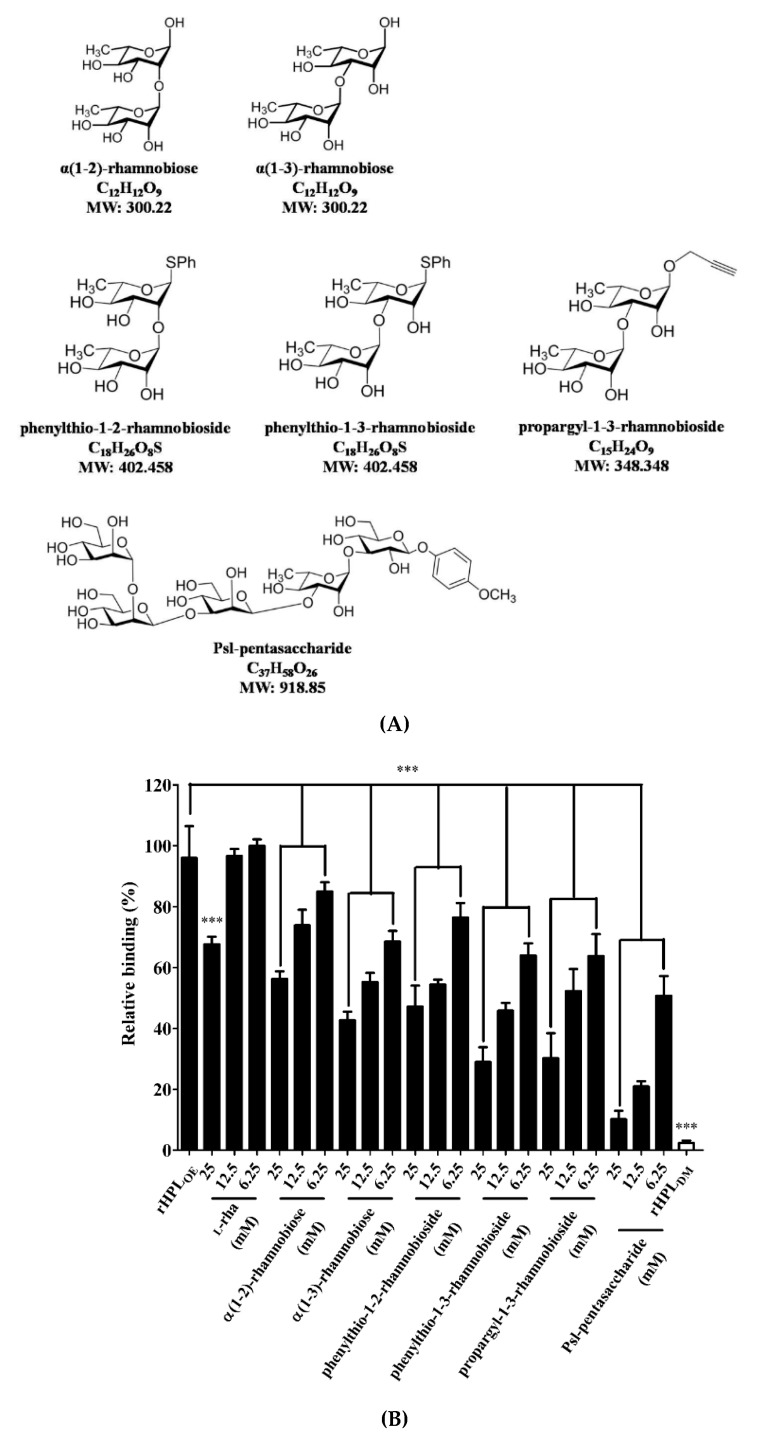
Structures of synthesized rhamnobioses and the rhamnose-containing Psl-pentasaccharide and inhibitory effects of thereof on the rHPL_OE_-biofilm interaction. (**A**) Structures of the synthesized rhamnosyl di- and pentasaccharides: α(1-2)-rhamnobiose, α(1-3)-rhamnobiose, phenylthio-1-2-rhamnobioside, phenylthio-1-3-rhamnobioside, propargyl-1-3-rhamnobioside and Psl-pentasaccharide. (**B**) The binding activity of rHPL_OE_ on the biofilm from *Pseudomonas aeruginosa* PA14 and the inhibitory effects of rhamnose or rhamnobiosides on this binding were determined by competitive ELISA. *** *p* < 0.001 versus the rHPL_OE_-only group.

**Figure 3 marinedrugs-17-00355-f003:**
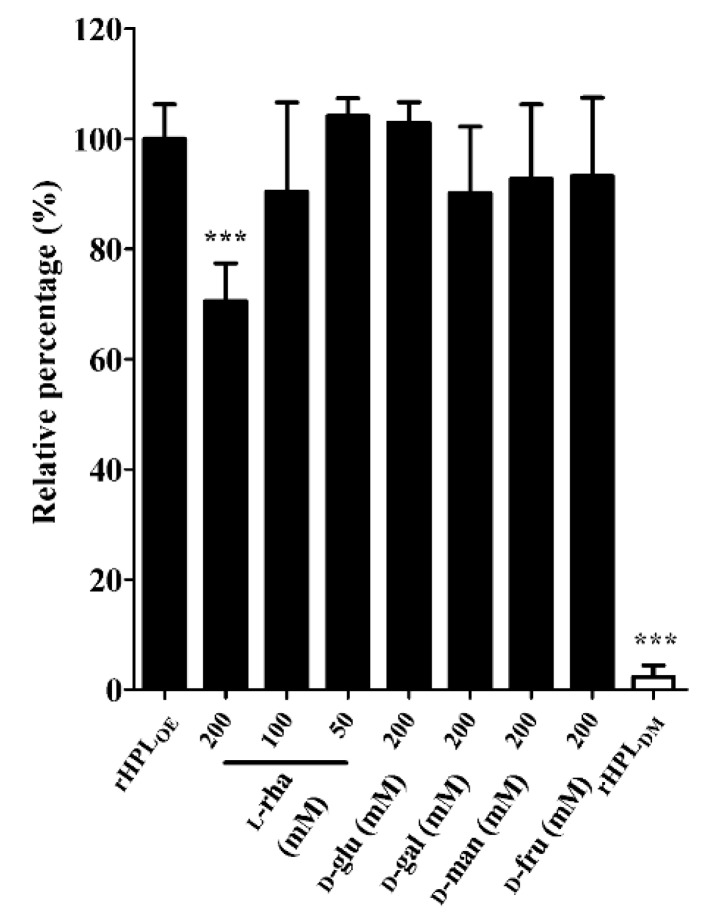
Inhibitory effects of monosaccharides on the rHPL_OE_-di-rhamnolipid interaction. The binding activity of rHPL_OE_ for di-rhamnolipid and inhibitory effects of monosaccharides on this binding was determined by competitive ELISA (competed with by l-rhamnose, d-glucose, d-galactose, d-mannose and d-fructose). *** *p* < 0.001 versus the rHPL_OE_-only group.

**Figure 4 marinedrugs-17-00355-f004:**
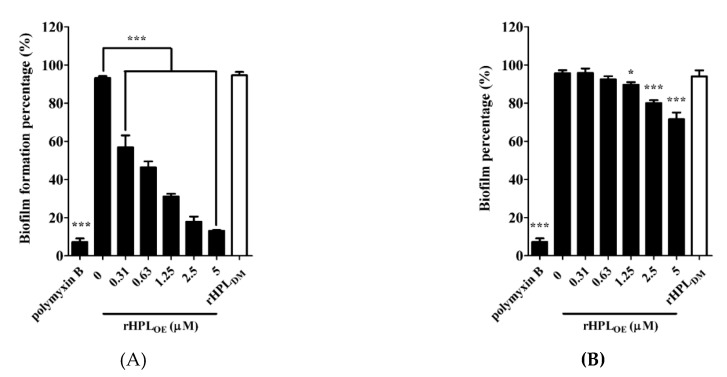
Inhibitory and dispersion effect on the biofilm of *P. aeruginosa* PA14 by rHPL_OE_. Quantification of crystal violet staining associated with (**A**) the biofilm of *P. aeruginosa* PA14 after treatment with rHPL_OE_ at the indicated concentration for 24 h and (**B**) after biofilm formation for 24 h and treatment with rHPL_OE_ at the indicated concentration for a further 24 h. rHPL_DM_ was applied as a negative control. The buffer-treated group was set as 100% (mock). * *p* < 0.05 and *** *p* < 0.001 versus the buffer-treated group.

**Figure 5 marinedrugs-17-00355-f005:**
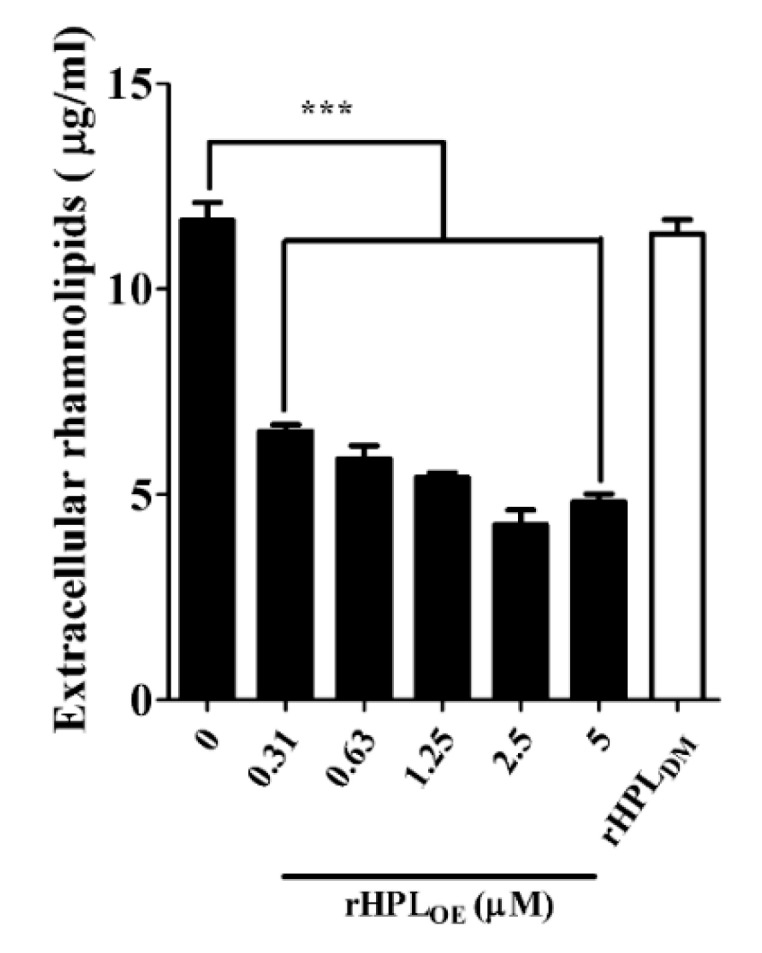
Down-regulation effect on extracellular rhamnolipids of *P. aeruginosa* PA14 by rHPL_OE_ or rHPL_DM_. Extracellular rhamnolipids of *P. aeruginosa* PA14 treated by rHPL_OE_ or rHPL_DM_ were measured by the chloroform–methyl blue method. *** *p* < 0.001 versus the buffer-treated group.

**Figure 6 marinedrugs-17-00355-f006:**
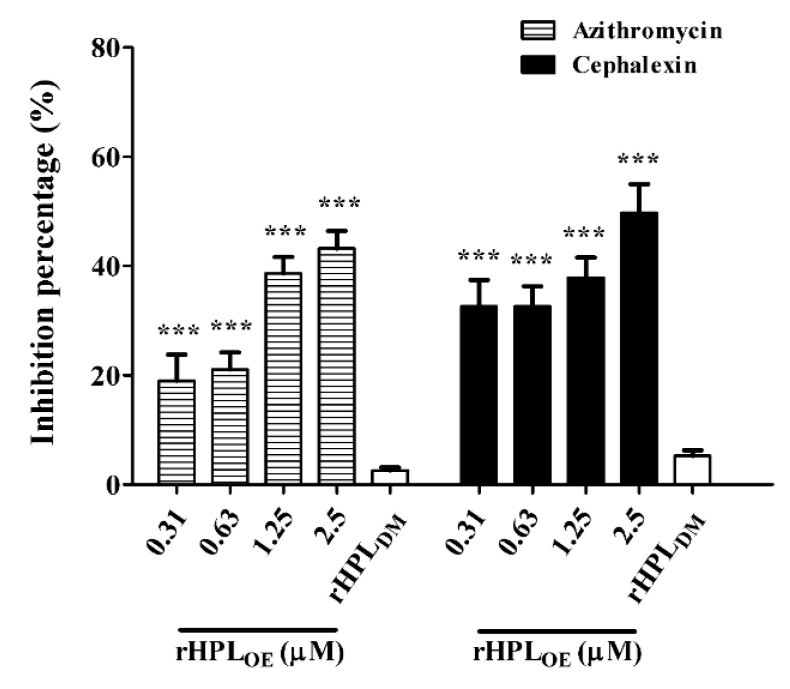
Synergistic effect of rHPL_OE_ with IC50 doses of azithromycin and cephalexin on *P. aeruginosa*. Quantity of the percentage of biofilm total protein inhibition (with respect to the antibiotic-only control) of *P. aeruginosa* PA14. *** *p* < 0.001 versus the control group.

**Figure 7 marinedrugs-17-00355-f007:**
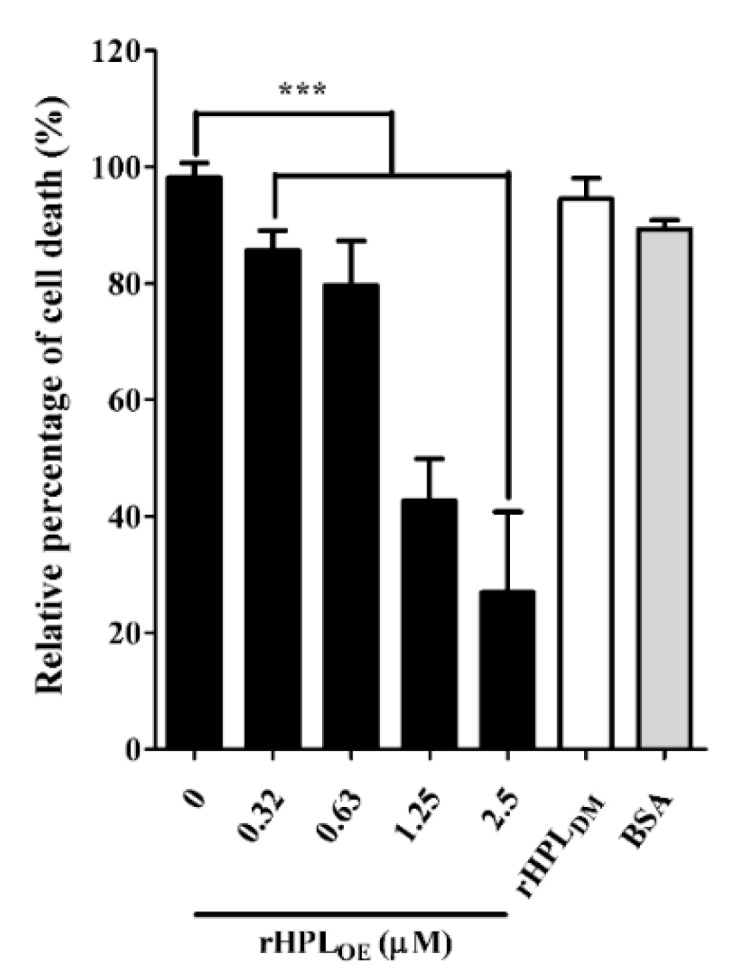
Inhibitory activities of rHPL_OE_ on *P. aeruginosa* PA14 infection of human A549 lung cells. The percent cell death was quantified by the alamarBlue cell viability assay and normalized to cells in the absence of rHPL_OE_, for which the cell death was set as 100%. *** *p* < 0.001 versus the control group.

**Figure 8 marinedrugs-17-00355-f008:**
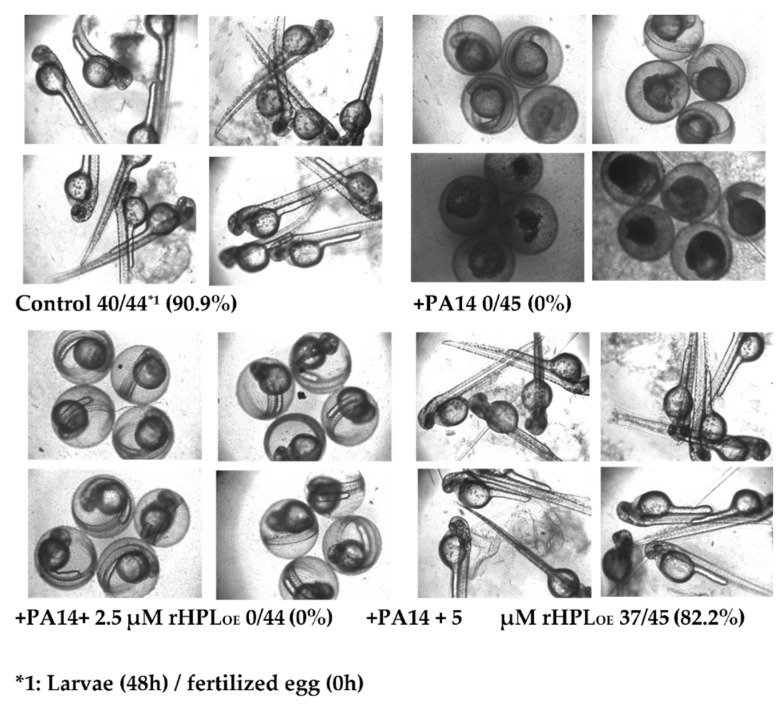
Inhibitory activity of rHPL_OE_ on *P. aeruginosa* PA14 infection of zebrafish embryos. Zebrafish embryos were infected with *P. aeruginosa* PA14 in the presence or absence of rHPL_OE_ and imaged after 48 h of infection. All of the pictures were taken with an inverted microscope at 40×.

**Table 1 marinedrugs-17-00355-t001:** Inhibition percentage (%) of QS-factors in rHPL_OE_ treated *P. aeruginosa* PA14.

	rHPL_OE_(μM)	0.31	0.63	1.25	2.5
QS-factors					
Activities of extracellular proteases		7 ± 1.4	4.8 ± 0.8	15.9 ± 2.2	12.4 ± 1.4
Pyoverdine		12.3 ± 0.2	8.2 ± 0.2	16.7 ± 0.2	21.2 ± 0.2
Pyocyanin		40.8 ± 13.9	56.1 ± 4	45.4 ± 9.2	40.5 ± 24.2

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
