# Peer review of "Rhamnose Binding Protein as an Anti-Bacterial Agent—Targeting Biofilm of Pseudomonas aeruginosa"

_marinedrugs, 2019, doi:10.3390/md17060355_

Reviewer 1 Report

The manuscript describes an alternative antimicrobial approach to contrast biofilm formation and virulence of Pseudomonas aeruginosa without affecting bacterial vitality. This approach is very interesting and now many reports in literature describe this novel strategy.

The manuscript is well written and rigorous in the scientific plan.

I suggest to improve the introduction section by adding some reports about new molecules of natural origin acting on biofilm of P. aeruginosa without affecting vitality of the cells avoiding the development of escape mutants:

Antimicrobial and Antibiofilm Activity and Machine Learning Classification Analysis of Essential Oils from Different Mediterranean Plants against Pseudomonas aeruginosa. Molecules. 2018 Feb 23;23(2). pii: E482. 

Anti-Biofilm Activities from Marine Cold Adapted Bacteria Against Staphylococci and Pseudomonas aeruginosa. Front Microbiol. 2015 Dec 14;6:1333. 

Author Response

Response to Reviewer 1 Comment

Point 1: I suggest to improve the introduction section by adding some reports about new molecules of natural origin acting on biofilm of P. aeruginosa without affecting vitality of the cells avoiding the development of escape mutants:

Antimicrobial and Antibiofilm Activity and Machine Learning Classification Analysis of Essential Oils from Different Mediterranean Plants against Pseudomonas aeruginosa. Molecules. 2018 Feb 23;23(2). pii: E482. 

Anti-Biofilm Activities from Marine Cold Adapted Bacteria Against Staphylococci and Pseudomonas aeruginosa. Front Microbiol. 2015 Dec 14;6:1333.  

Response 1:

According this suggestion, we add some contents in the introduction section and the reference section as follows.

Line #66-69

There already exist many different anti-biofilm agents from various resources. For example, cultured broth from certain marine cold adapted bacteria destabilized biofilm of Pseudomonas aeruginosa [17] and some essential oils from Mediterranean plants or selected exopolysaccharide from marine bacteria acts as anti-QS factors to inhibit biofilm formation of Pseudomonas aeruginosa [18,19].

Line #610-619

17.           Papa, R.; Selan, L.; Parrilli, E.; Tilotta, M.; Sannino, F.; Feller, G.; Tutino, M.L.; Artini, M. Anti-Biofilm Activities from Marine Cold Adapted Bacteria Against Staphylococci and Pseudomonas aeruginosa. Frontiers in microbiology 2015, 6, 1333, doi:10.3389/fmicb.2015.01333.

18.           Artini, M.; Patsilinakos, A.; Papa, R.; Bozovic, M.; Sabatino, M.; Garzoli, S.; Vrenna, G.; Tilotta, M.; Pepi, F.; Ragno, R., et al. Antimicrobial and Antibiofilm Activity and Machine Learning Classification Analysis of Essential Oils from Different Mediterranean Plants against Pseudomonas aeruginosa. Molecules 2018, 23, doi:10.3390/molecules23020482.

19.           Wu, S.; Liu, G.; Jin, W.; Xiu, P.; Sun, C. Antibiofilm and Anti-Infection of a Marine Bacterial Exopolysaccharide Against Pseudomonas aeruginosa. Frontiers in microbiology 2016, 7, 102, doi:10.3389/fmicb.2016.00102.

Reviewer 2 Report

The manuscript by Fu et al. entitled "Rhamnose Binding Protein as Anti-Bacterial Agent - Targeting Biofilm of Pseudomonas aeruginosa" presents the results of interesting research on the control of bacterial biofilm by a protein modeled on the natural hemolymph lectin of marine organism Tachypleus tridentatus. The authors carried out extensive research in an attempt to establish the mechanisms of anti-biofilm action of this product on the Pseudomonas aeruginosa biofilm model. I do not have serious criticisms about the goals and the research done.

However, numerous corrections should be introduced - see comments below.

Introduction section, generally speaking, pretty well highlights the problem of alternative methods of controlling biofilm and justifies the conduct of this study.
However, I draw the authors' attention to the often too long sentences; editorial errors (unnecessary italics or lack of italics); to a few sentences that are unclear (incomprehensible in their content): lines 75-77, 85-86, 91-95.

Results

This section presents in a clear way the results of comprehensive in vitro and in vivo studies, that enable their analysis. However,

- The figures are not very legible - you have to enlarge them.

- Legends to Figures should be somewhat simplified since the details of the experiments are described in detail in the M&M section.

Discussion

This section contains part of the information which are also in the Introduction and contains too much repetition of the result descriptions. This part of the MS should be more "synthetic" and therefore shortened.

- I suggest the authors need to extract a summary / conclusion subsection. Clearly, this part of MS is currently missing. In view of the wide range of research and received valuable and quite numerous results - this will allow in a more friendly way to emphasize the achievements of the authors.

Material and methods

- Line 398 -please explain .."Pseudomonas aeruginosa PA14 (serotype O10 or O19) was "...

- Line 423 - "4.3. Biofilm formation and extraction" - The content description of this subsection is unclear. You have to organize it.

- Line 438 - a dot separating two sentences is needed.

- Line 443 - the title of this subsection is unclear.

- Lines 535-541 - subsection "4.12. Synergistic effect of rHPLOE with antibiotics" and results of this study (subsection "2.8. Combination treatment with rHPLOE improved efficacies of antibiotics on P. aeruginosa with preformed biofilms")  -  to be allow to talk about the synergistic effect of two factors need to be carried out a classic "checkerboard" test, plot of isobologram and determine the index FICI. Can the authors comment on this?

Author Response

Response to Reviewer 2 Comments

Point 1:

Introduction section, generally speaking, pretty well highlights the problem of alternative methods of controlling biofilm and justifies the conduct of this study.
However, I draw the authors' attention to the often too long sentences; editorial errors (unnecessary italics or lack of italics); to a few sentences that are unclear (incomprehensible in their content): lines 75-77, 85-86, 91-95.

Response 1:

We correct some errors and re-write sentences (lines 75-77, 85-86, 91-95) as follows.

Line# 56
N-(3-oxododecanoyl)-
L-homoserinelactone” to “N-(3-oxododecanoyl)-L-homoserine lactone”

Line# 71
P. aeruginosa” to P. aeruginosa”

Line# 87
 “O-antigen” to” O-antigen”

Line# 79-81(75-77)

Since the mechanism of di-rhamnolipid on influencing biofilm development remains unclear, anti-biofilm reagents targeting di-rhamnolipid might be a potential solution for a novel anti-biofilm approach.

is replaced by:

The influence of di-rhamnolipid to biofilm development was observed, but the interaction of di-rhamnolipid with other QS- factors remains unclear. To develop anti-biofilm reagents directly targeting di-rhamnolipid might be a novel anti-biofilm approach.

Line# 88-90 (85-86)

Hence RBPs are naturally related to the innate immune system in animals as a pathogen recognition element.

is replaced by:

 “The tissue specificity of expression and ability to interact with bacteria indicates that RBPs are naturally related to the innate immune system in animals as a pathogen recognition element.

Line#95-98 (91-95)

On the other hand, other RBLs bind to rhamnose but also mannose or galactose. This broad specificity indicated that RBLs cannot specifically target to the rhamnose-containing components in biofilms. The high specificity to rhamnose and rhamnobiose [30] makes rHPL a prospective candidate to bind and interrupt biofilm development of P. aeruginosa by recognizing rhamnose.”

is replaced by:

 “Unlike other RBLs, rHPL only binds to L-rhamnose and rhamnobiose but not to galactose or mannose [30]. This high substrate specificity makes rHPL a prospective candidate to bind rhamnose-containing components in biofilm of P. aeruginosa and examine the biological functions of these bindings.”

Point 2:

Results

This section presents in a clear way the results of comprehensive in vitro and in vivo studies, that enable their analysis. However,

- The figures are not very legible - you have to enlarge them.

- Legends to Figures should be somewhat simplified since the details of the experiments are described in detail in the M&M section.

Response 2:

We enlarge all figures with resolution at 300 dpi.

All legends are simplified as follow:

Line #138-141

 “Figure 1. Inhibitory effects of monosaccharides on rHPLOE-biofilm interaction. The binding activity of rHPLOE to the PA14 biofilm and inhibitory effects of monosaccharides or alginate on this binding were determined by competitive ELISA. rHPLDM was applied as a negative control. ***P<0.001 versus the rHPLOE only group (positive control).

Line# 154-160

 “Figure 2. Structures of synthesized rhamnobioses and the rhamnose-containing Psl-pentasaccharide and inhibitory effects of thereof on rHPLOE-biofilm interaction. (A) Structures of the synthesized rhamnosyl di- and pentasaccharides: α(1-2)-rhamnobiose, α(1-3)-rhamnobiose, phenylthio-1-2-rhamnobioside, phenylthio-1-3-rhamnobioside, propargyl-1-3-rhamnobioside and Psl-pentasaccharide. (B) The binding activity of rHPLOE on the biofilm from P. aeruginosa PA14 and the inhibitory effects of rhamnose or rhamnobiosides on this binding were determined by competitive ELISA. ***P<0.001 versus the rHPLOE-only group.

Line# 174-177

 “Figure 3. Inhibitory effects of monosaccharides on rHPLOE-di-rhamnolipid interaction. The binding activity of rHPLOE for di-rhamnolipid and inhibitory effects of monosaccharides on this binding were determined by competitive ELISA (competed with by L-rhamnose, D-glucose, D-galactose, D-mannose and D-fructose). ***P<0.001 versus the rHPLOE-only group.

Line# 195-200

 “Figure 4. Inhibitory and dispersion effect on the biofilm of P. aeruginosa PA14 by rHPLOE. Quantification of crystal violet staining associated with (A) the biofilm of P. aeruginosa PA14 after treatment with rHPLOE at the indicated concentration for 24 h and (B) after biofilm formation for 24 h and treatment with rHPLOE at the indicated concentration for a further 24 h. rHPLDM was applied as a negative control. The buffer-treated group was set as 100% (mock). *P< 0.05 and ***P< 0.001 versus the buffer-treated group.

Line# 216-218

 “Figure 5. Down-regulation effect on extracellular rhamnolipids of P. aeruginosa PA14 by rHPLOE or rHPLDM. Extracellular rhamnolipids of P. aeruginosa PA14 treated by rHPLOE or rHPLDM were measured by the chloroform-methyl blue method. ***P<0.001 versus the buffer-treated group.

Line# 252-254

 “Figure 6. Synergistic effect of rHPLOE with IC50 doses of azithromycin and cephalexin on P. aeruginosa. Quantity of the percentage of biofilm total protein inhibition (with respect to the antibiotic-only control) of P. aeruginosa PA14. ***P<0.001 versus the control group.

Line# 269-272

 “Figure 7. Inhibitory activities of rHPLOE on P. aeruginosa PA14 infection of human A549 lung cells. The percent cell death was quantified by the alamarBlue cell viability assay and normalized to cells in the absence of rHPLOE, for which the cell death was set as 100%. *P< 0.05, **P< 0.01, and ***P< 0.001 versus the control group.

Line# 285-287

 “Figure 8. Inhibitory activity of rHPLOE on P. aeruginosa PA14 infection of zebrafish embryos. Zebrafish embryos were infected with P. aeruginosa PA14 in the presence or absence of rHPLOE and imaged after 48 h of infection. All of the pictures were taken with an inverted microscope at 40X.

Point 3:

Discussion

This section contains part of the information which are also in the Introduction and contains too much repetition of the result descriptions. This part of the MS should be more "synthetic" and therefore shortened.

- I suggest the authors need to extract a summary / conclusion subsection. Clearly, this part of MS is currently missing. In view of the wide range of research and received valuable and quite numerous results - this will allow in a more friendly way to emphasize the achievements of the authors.

Response 3:

The Discussion section is re-arranged and separated to Discussion and Conclusion. Some repetitions and redundant sentences are removed as follow.

Line# 293-355

3. Discussion

To study the carbohydrate components of the biofilm, carbohydrate-binding proteins, such as plant lectins, have been widely applied. Previously many different neutral carbohydrates including N-acetyl-D-glucosamine, N-acetyl-D-galactosamine, D-glucose, and D-mannose were identified in the exopolymeric matrix of the biofilm [38]. Lectins with various specificities show interaction with the carbohydrate components of biofilms. For example, concanavalin A from Canavalia ensiformis seeds is specific for D-glucose and D-mannose [39], and lectin from Triticum vulgaris germs is specific for N-acetyl-D-glucosamine and sialic acid [40]. L-Rhamnose as a unique sugar in bacteria and plants is a common component of the cell wall and capsule of many pathogenic bacteria including Gram-negative P. aeruginosa [41], Salmonella typhimurium [42], Vibrio cholera [43,44] as well as Mycobacterium tuberculosis [45]. P. aeruginosa produces at least three distinct exopolysaccharides that contribute to biofilm development and architecture: alginate, Pel, and Psl [46]. Alginate consists of only uronic acids, Pel is a glucose-rich polysaccharide, and Psl consists of a repeating pentasaccharide containing D-mannose, D-glucose and L-rhamnose [47]. Psl serves as a primary structural scaffold for biofilm development. In addition, Psl is involved in early stages of biofilm formation and promoting cell–to-cell interactions [48,49]. Another rhamnose-containing component in the biofilm was the QS-factor di-rhamnolipid. A mutant strain (P. aeruginosa PAO1C-rhlAB) that could not produce rhamnolipids lost its swarming activity, which agreed with our observation in rHPLOE-treated PA14 [50]. rHPLOE bound to both Psl and di-rhamnolipid through targeting rhamnose moiety and synthetic rhamnobiosides-competitive ELISA showed that rHPLOE preferred to bind with α(1-3)-rhamnobiose rather than α(1-2)-rhamnobiose. The linkage of the rhamnose of Psl was reported to be α(1-3) [47] while that of di-rhamnolipid was α(1-2) [51], perhaps favoring that rHPLOE binding to rhamnose component in biofilm was governed by recognition of unique structure feature of rhamnosyl moiety. Although rHPLOE binding to Psl still required more abundant glycan for detail characterization, rHPLOE binding to di-rhamnolipid down-regulated the expression of di-rhamnolipid and QS-factors of P. aeruginosa and further inhibited biofilm formation.

The most important finding of this study was that the binding of rHPLOE to the P. aeruginosa PA14 biofilm inhibited biofilm development and disrupted the mature biofilm. Our data clearly showed that rHPLOE caused these bio-effects by interacting with components in the biofilm including structural polysaccharides or di-rhamnolipid, largely due to down-regulation of QS-factors including di-rhamnolipid, pyocyanin, pyoverdine, and extracellular proteases by interrupting the functions of di-rhamnolipid.

Based on the biofilm dispersion activity of rHPLOE, we proposed that rHPLOE possessed synergistic effects with antibiotics on P. aeruginosa. Two antibiotics commonly applied for treating P. aeruginosa infections, azithromycin, a hydrophobic azalide that kills bacteria by decreasing protein production, and cephalexin, a hydrophilic beta-lactam that kills bacteria by inhibiting cell wall synthesis, were used in this study in combination with rHPLOE. The results showed that rHPLOE significantly improved the bactericidal activity of both antibiotics, strongly suggesting that rHPLOE was useful as a biofilm dispersion reagent for deconstructing the biofilm and improving the activities of antibiotics.

Many studies reported that P. aeruginosa infection can be inhibited by down-regulating QS-factors. Importantly, these anti-QS reagents (allicin, triazolyldihydrofuranone, and baicalin hydrate) are effective against multidrug-resistant P. aeruginosa [33-35]. Studies in the past decade revealed that these anti-QS reagents also inhibited many human infections caused by biofilm-producing bacteria [52,53]. This fact is important for fighting human pathogenic bacteria, as biofilms are found to be involved in over 80% of microbial infections in humans [54]. Since rHPLOE could reduce the levels of pyoverdine and pyocyanin, we speculate that rHPLOE might also inhibit P. aeruginosa infection. It was found that rHPLOE significantly reduced the cytotoxicity towards A549 cells and neutralized toxicity (leading to development retardation or death) of the zebrafish embryo caused by P. aeruginosa PA14.

4. Conclusion

Binding and down-regulation of di-rhamnolipid has not been attracted much attention so far. In this study, we optimized the production of rHPL in E. coli using chaperone coexpression. rHPLOE, a highly specific rhamnose binding protein, bound to not only bacterial cells and PAMPs [29] but also extracted cell-free biofilm from P. aeruginosa. In addition, such interaction inhibited biofilm formation and dispersed mature biofilm through down-regulating secretion of di-rhamnolipid in biofilm and further down-regulating other QS-factors including extracellular proteases, pyoverdine, and pyocyanin. As a biofilm dispersion reagent, rHPLOE increased the antibiotic activity against P. aeruginosa PA14 with pre-formed biofilm. Therefore, rHPLOE promised to be effective anti-biofilm agent for combination therapy. At cellular and animal levels, rHPLOE inhibited the infection and toxicity of P. aeruginosa PA14 towards human lung epithelial cells and zebrafish embryos. These results indicated that rHPLOE served as a novel anti-biofilm agent by targeting rhamnose-containing components in biofilm, which in turn linked glycan-recognition to novel anti-biofilm strategies against pathogenic bacteria.

Point 4:

Material and methods

- Line 398 -please explain .."Pseudomonas aeruginosa PA14 (serotype O10 or O19) was "...

Response 4:

Our information of this strain was that Pseudomonas aeruginosa PA14 might belong to serotype O10 or O19. After searching more references, we confirm that Pseudomonas aeruginosa PA14 was classified to serotype O19 [1]. We correct this error as follow

Line# 360

Pseudomonas aeruginosa PA14 (serotype O10 or O19) “

is replaced by:

Pseudomonas aeruginosa PA14 (serotype O19)”

Point 5:

Material and methods

- Line 423 - "4.3. Biofilm formation and extraction" - The content description of this subsection is unclear. You have to organize it.

Response 5:

This subsection is re-organized as follow:

Bacterial biofilm extraction was as described by Chibaet et al.[55]. P. aeruginosa PA14 grown on LB plates was inoculated in LB medium and incubated overnight at 37°C with 250 rpm circular shaking. A portion of the overnight culture was 1,000-fold diluted in 10 mL LB medium and incubated at 37°C to an A600nm of 1. Bacterial cells with biofilm were harvested from incubated solution by centrifugation at 8,000 x g for 10 min at 25°C (KUBOTA, Osaka, Japan). The harvested pellet was re-suspended with 1 mL of 1.5 M NaCl to extract cell-free biofilm component. The suspensions were centrifuged at 5,000 x g for 10 min at 25°C (KUBOTA, Osaka, Japan), and the supernatants containing the biofilm fraction were collected.

Point 6:

Material and methods

- Line 438 - a dot separating two sentences is needed.

Response 6:

A dot is added between two sentences at Line# 400

Point 7:

Material and methods

- Line 443 - the title of this subsection is unclear.

Response 7:

Line# 405

4.5. Biofilm or di-rhamnolipid binding activity measurement

is replaced by

4.5. Binding activity of rHPLOE to biofilm or di-rhamnolipid

Point 8:

Material and methods

- Lines 535-541 - subsection "4.12. Synergistic effect of rHPLOE with antibiotics" and results of this study (subsection "2.8. Combination treatment with rHPLOE improved efficacies of antibiotics on P. aeruginosa with preformed biofilms")  -  to be allow to talk about the synergistic effect of two factors need to be carried out a classic "checkerboard" test, plot of isobologram and determine the index FICI. Can the authors comment on this?

Response 8:

Before we did the combination test, we had tested the inhibition activity of azithromycin or cephalexin to P. aeruginosa PA14 with pre-formed biofilm with same method. The IC50 of these two antibiotics were determined: 27.3±1.4 μg/mL of azithromycin and 27.7±0.9 μg/mL of cephalexin. We add this information to Line# 246-248 as follow

“The IC50 of azithromycin and cephalexin to P. aeruginosa PA14 with pre-formed biofilm was 27.3±1.4 and 27.7±0.9 μg/mL, respectively.”

Based on this data, we designed combination test of rHPLOE with identical concentrations of antibiotics (25 μg/mL) follow method of Das et al. [2] and showed our data follow their way.

We add this information to Line# 498 as follow

“The experiment design was according to the method described by Das et al. [63].”

Reference:

1.         Hao, Y.; Murphy, K.; Lo, R.Y.; Khursigara, C.M.; Lam, J.S. Single-Nucleotide Polymorphisms Found in the migA and wbpX Glycosyltransferase Genes Account for the Intrinsic Lipopolysaccharide Defects Exhibited by Pseudomonas aeruginosa PA14. Journal of bacteriology 2015, 197, 2780-2791, doi:10.1128/JB.00337-15.

2.         Das, M.C.; Sandhu, P.; Gupta, P.; Rudrapaul, P.; De, U.C.; Tribedi, P.; Akhter, Y.; Bhattacharjee, S. Attenuation of Pseudomonas aeruginosa biofilm formation by Vitexin: A combinatorial study with azithromycin and gentamicin. Scientific reports 2016, 6, 23347, doi:10.1038/srep23347.